# How, when, and why do inter-organisational collaborations in healthcare work? A realist evaluation

Justin Avery Aunger[1,2]*, Ross Millar[2], Anne Marie Rafferty[3], Russell Mannion[2], Joanne Greenhalgh[4], Deborah Faulks[5], Hugh McLeod[6]

1 School of Health Sciences, Surrey Research Park, Surrey, United Kingdom, 2 Health Services Management Centre, Park House, University of Birmingham, Birmingham, United Kingdom, 3 Florence Nightingale Faculty of Nursing, Midwifery and Palliative Care, King's College London, London, United Kingdom, 4 Sociology and Social Policy Department, University of Leeds, Leeds, United Kingdom, 5 Engaging Communities Solutions, Stafford, United Kingdom, 6 Population Health Sciences, University of Bristol & NIHR Applied Research Collaboration West, Bristol, United Kingdom

* j.aunger@surrey.ac.uk

**Data Availability Statement:** Data cannot be shared publicly because of anonymity requirements of the involved participants and because the data originate in unanonymised

## Abstract

### Background

Inter-organisational collaborations (IOCs) in healthcare have been viewed as an effective approach to performance improvement. However, there remain gaps in our understanding of *what* helps IOCs function, as well as *how* and *why* contextual elements affect their implementation. A realist review of evidence drawing on 86 sources has sought to elicit and refine context-mechanism-outcome configurations (CMOCs) to understand and refine these phenomena, yet further understanding can be gained from interviewing those involved in developing IOCs.

### Methods

We used a realist evaluation methodology, adopting prior realist synthesis findings as a theoretical framework that we sought to refine. We drew on 32 interviews taking place between January 2020 and May 2021 with 29 stakeholders comprising IOC case studies, service users, as well as regulatory perspectives in England. Using a retroductive analysis approach, we aimed to test CMOCs against these data to explore whether previously identified mechanisms, CMOCs, and causal links between them were affirmed, refuted, or revised, and refine our explanations of how and why interorganisational collaborations are successful.

### Results

Most of our prior CMOCs and their underlying mechanisms were supported in the interview findings with a diverse range of evidence. Leadership behaviours, including showing vulnerability and persuasiveness, acted to shape the core mechanisms of collaborative functioning. These included our prior mechanisms of trust, faith, and confidence, which were largely

qualitative transcripts which would identify these participants. Data are available from the University of Birmingham Ethics Committee (contact via researchgovernance@contacts.bham.ac.uk) for researchers who meet the criteria for access to confidential data.

**Funding:** All authors were supported by a grant from the NIHR Health Services and Delivery Research fund with grant number NIHR127430 (https://fundingawards.nihr.ac.uk/award/NIHR127430). The funders had no role in study design, data collection and analysis, decision to publish, or preparation of the manuscript.

**Competing interests:** The authors have declared that no competing interests exist.

ratified with minor refinements. Action statements were formulated, translating theoretical findings into practical guidance.

## Conclusion

As the fifth stage in a larger project, our refined theory provides a comprehensive understanding of the causal chain leading to effective collaborative inter-organisational relationships. These findings and recommendations can support implementation of IOCs in the UK and elsewhere. Future research should translate these findings into further practical guidance for implementers, researchers, and policymakers.

## Introduction

Inter-organisational collaborations (IOCs) have been promoted for decades internationally as an approach to organisational improvement and turnaround in healthcare [1, 2]. Such collaborations, defined by the sharing of a common underlying commitment to joint working across organisations to attain benefits that would not otherwise be achieved by working alone [3], encompass a range of organisational structures. These include buddying [4], provider alliances [5], Accountable Care Organisations [6], and mergers [7], which can sit on a spectrum from low to high degrees of integration [8, 9]. Such arrangements can exist across sectors (e.g., both health and social care) and within the same sector (e.g., only involving acute care).

Current examples of ambitious IOCs operating across sectors in England are Integrated Care Systems (ICSs). Building on testing new care models termed 'Vanguards' [10], these are a mandated form of geographically-based cross-sector IOC. ICSs have been implemented across England since April 2021 and are due to become statutory later in 2022 [11, 12]. As such, this research takes place in a policy context characterised by decades of competition as the key mechanism to drive performance and a current shift towards collaboration. However, mandating collaboration is likely to exert a significant influence over the process and may complicate relationships that have already been built between healthcare providers, the impact of which is currently not well-understood [13].

The extent to which IOCs deliver the expected benefits to care quality, safety, and efficiency, is contested [13–16]. Implementing IOCs means additional complexity in the ways in which people work, and involves significant time, financial resource, and effort, before any benefits may be realised [3, 17]. There remain gaps in our understanding of *what* is key in helping IOCs function, as well as *how* and *why* various contextual factors affect their implementation. Several literature reviews [16, 18–20] and qualitative evaluations [21–23] have sought to explore the success factors and barriers underlying various forms of IOCs, but they have not explored the mechanisms which drive change in IOCs, nor link functioning to *how* collaborative behaviour is manifested. We identified an initial programme theory that articulated how collaborations are intended to work [15] and refined this theory by drawing on case study literature [3]. A core assumption of our theory is that there are simultaneous, competing organisational drivers to compete and collaborate and these shape the mechanisms through which collaboration occurs and the outcomes produced [15]. Our theory also hypothesises that collaborative behaviour occurs in a context of better collaborative functioning, where trust, faith, and confidence are maximised. This, in turn, enables performance improvements to be attained.

## Rationale

Realist methodology is ideal for unpacking the 'black box' of how interventions work, and realist evaluations that have sought to better understand *how* and *why* such IOCs work are already present in the literature [24–27]. Most of these have focused on one type of IOC, however, assessing multiple types of IOC in a single realist evaluation may assist in identifying which underlying mechanisms are shared across IOC interventions and may provide a better understanding of *how* and *why* they work across different contexts.

In the present research, we conducted a realist evaluation drawing upon multiple case studies and stakeholder analyses with interviewees familiar with a range of IOCs in England. We sought a better understanding of *how* and *why* IOCs work, as well as *whom* they benefit. This was achieved by gathering insights derived from interviews with individuals working in different IOCs at various levels, in collaborating organisations, as well as regulators.

## Developing an understanding of IOC functioning

While realist evaluations usually form a standalone project that draw on a range of data sources, our approach is similar to that of Jagosh et al. (2015). These authors performed a realist synthesis of literature about participatory research programmes and later followed this with a realist evaluation informed by their earlier synthesis [27]. To date, our existing theoretical framework has been developed over the course of three stages: 1) an initial realist theory, drawing on organisational strategic documents and typologies of collaboration [8], 2) a realist review drawing on case studies, reviews, and theories of collaboration, [15], and 3) a refinement stage drawing purely on an expanded range of case study literature [3]. The present paper reports the results of Stages 4 and 5 of the research (Fig 1). It involves 'testing' and refining existing Context-Mechanism-Outcome Configurations (CMOCs) and the wider connections between them against the findings in the realist evaluation.

Our existing realist theory proposes that IOCs ideally (from the point of view of attempting to successfully collaborate) move from competitive behaviour towards collaborative behaviour. This assumes that a shift from individualistic organisational behaviour to collaborative behaviour depends on preceding mechanisms of collaborative functioning becoming optimised. Key to this process is *the building of trust*, which can increase *risk tolerance*. Greater risk tolerance enables organisations to act selflessly in the collaboration (i.e. take on additional risk) without fear of reprisal or being taken advantage of [3]. Likewise, building of *faith* is similarly important, where faith (the perception that the collaboration is virtuous and worth working on) also drives desire to collaborate. As such, these dual motives of *trust* and *faith* both need to be high to drive collaboration (Fig 2). In some mandated types of collaboration (e.g., mandated buddying) as well as those that have a higher degree of integration (i.e., mergers), we posit that *confidence* can 'stand in' for genuine interorganisational trust. Confidence, as a mechanism, refers to an understanding that a partner will act collaboratively because they are essentially contractually obligated to do so. This raises the risk tolerance of the partner in a similar manner to trust, allowing them to also act more collaboratively. As such, confidence arises from the formalisation of collaborative behaviour through contractual obligation. These mechanisms and others are defined in Table 2.

There are many other mechanisms outlined within collaborative functioning. One example is conflict resolution and accountability, which modulates conflict further down the causal chain. In this relationship, appropriate conflict resolution mechanisms can lessen the negative impact of conflict on trust and faith (Fig 2). Likewise, a stronger 'perception of progress' can drive increases in faith, and vice versa. Similarly, the perception of task complexity, or how complex actors perceive the collaboration to be to implement, also shapes faith. In this case, a

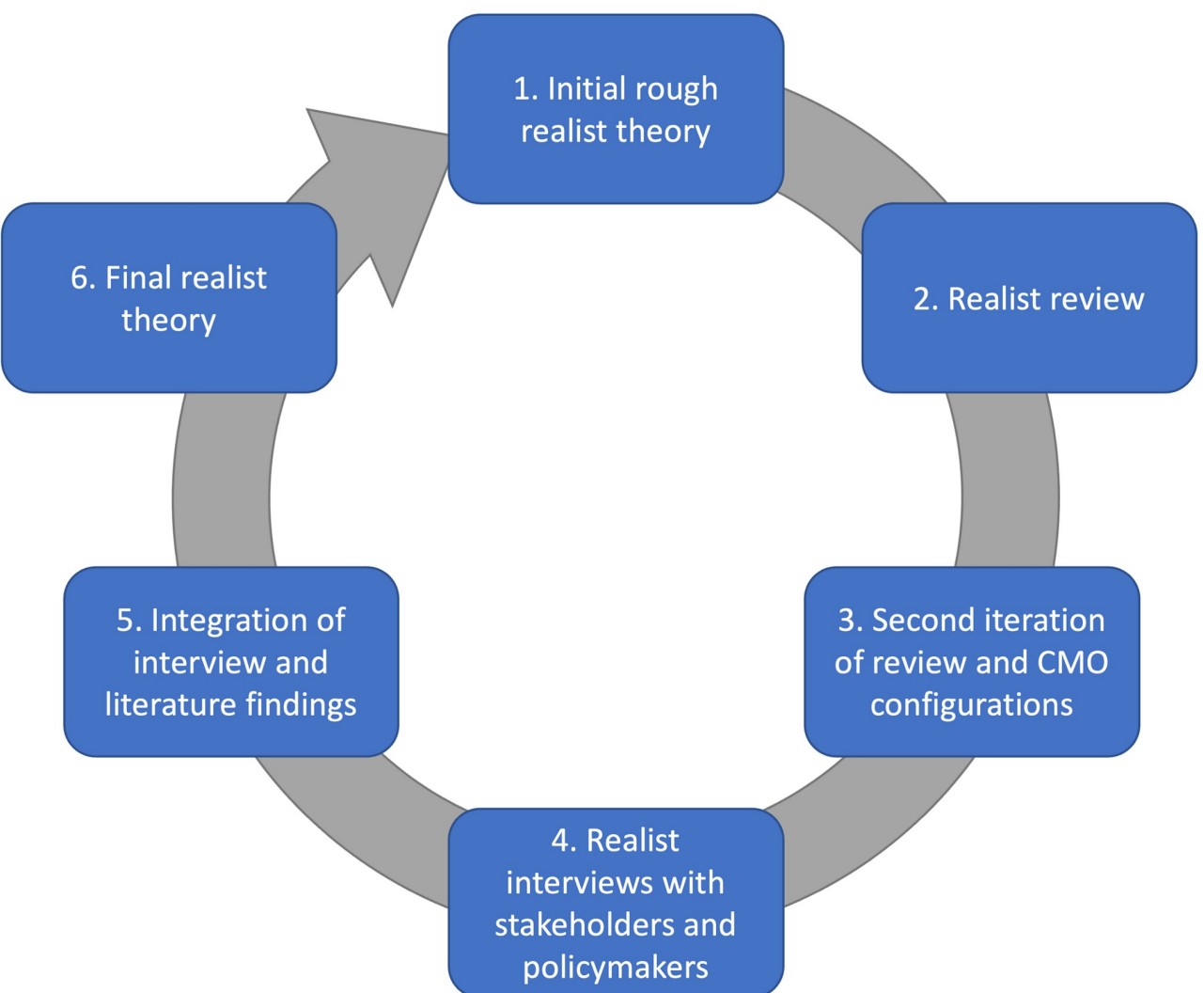

**Fig 1. Stages of this realist project.** This paper focused on stages 4 and 5 [3]. Originally published by Springer Nature (BMC Health Services Research). Reproduced with permission from the copyright holders.

high perception of complexity reduces faith, as people start to think the IOC might be too much work to realistically achieve. Some of these key relationships are outlined in Fig 2. These mechanisms are shaped by several contextual features. In the figure, the regulatory environment, resource constraints, and number of involved organisations all play a role in determining task complexity, and thereby faith. Further exposition of the theory and all the CMOCs we go on to refine in this paper are provided in our prior realist review [15] and refined theory papers [3].

## Evaluation objectives

We conducted a realist evaluation to further test our refined programme theory by exploring the experiences of a range of stakeholders across several examples of IOCs in England. Our primary objective was to test the mechanisms of collaborative functioning and CMOC derived from literature against the stakeholders' views. Thus, we aimed to explore whether previously

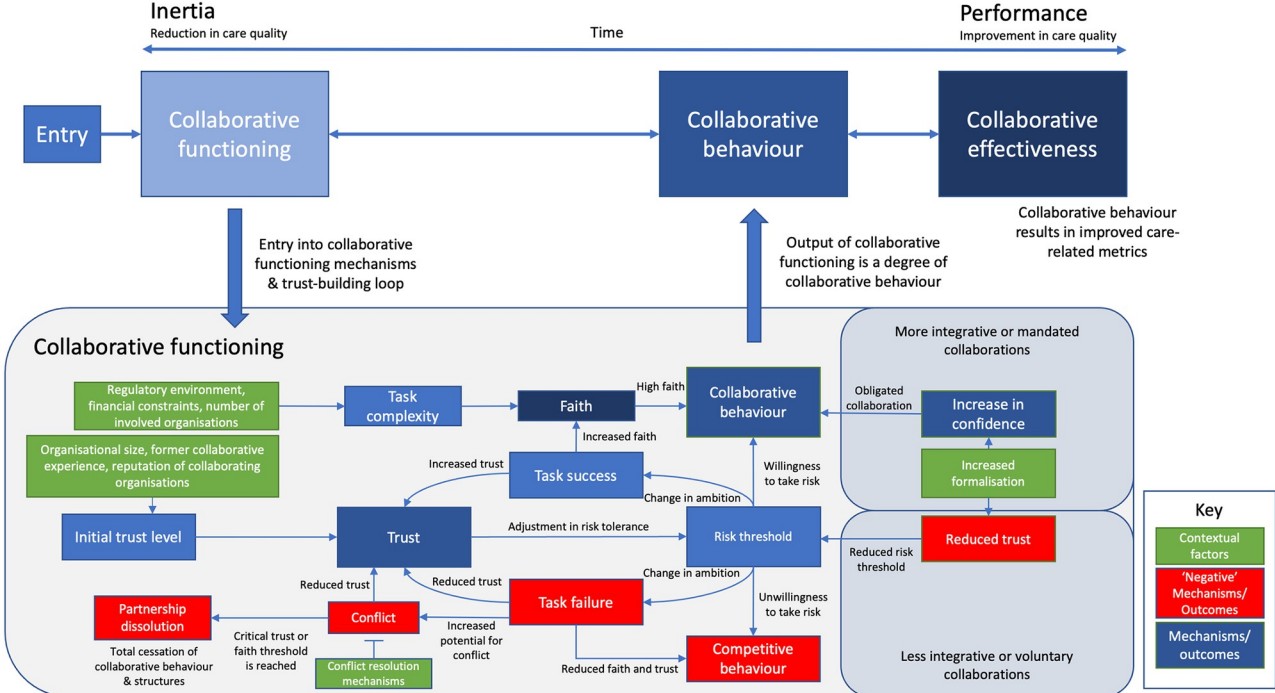

**Fig 2. Existing programme theory/theoretical framework based on Aunger, Millar & Greenhalgh (2021) [3].** Originally published by Springer Nature (BMC Health Services Research). Reproduced with permission from the copyright holders.

identified mechanisms, CMOCs, and causal links between them were affirmed, refuted or revised, and refine our understanding of how IOCs work, in which circumstances and why. We intended to produce a refined realist programme theory rooted in both literature and practice with a view towards practical use in the future by policymakers, researchers, and those implementing IOCs.

## Methods

### Realist evaluation methodology

A realist evaluation methodology was adopted for this study, while the wider project drew on a combination of realist synthesis and evaluation [28]. Realist evaluation was developed in the 1990s in seminal work by Pawson and Tilley (1997) for addressing 'what works, for whom, and why' in a range of interventions [28]. Underpinning this approach is a realist ontology, which assumes that the interventions are 'real' structures and systems that exist independently in the world. Likewise, it subscribes to the notion of 'generative causation', meaning that the intervention, delivered in different contexts, can trigger mechanisms to produce various outcomes [29]. These mechanisms, in realist terms, comprise a combination of resources introduced by the programme and changes to actor reasoning or behaviour caused by the programme [30]. Similarly, contexts, have generally been defined both as observable features (i.e., 'things') that trigger mechanisms, or as relational and dynamic features operating in an emergent manner within a social system [31]. In our analysis, contexts can adopt both forms. How contexts change the activation of mechanisms to produce certain outcomes can be expressed in terms of CMOCs. Eliciting and testing these CMOCs to develop an explanation

of how programmes work, in which circumstances, and why, is the main purpose of realist evaluations.

In the present realist evaluation, our prior programme theory [3] was adopted as a middle-range theory against which interview data were 'tested'. This means that we explored whether and how interview findings affirmed prior CMOCs, if refinements to existing ones were required, or to which degree identified CMOCs were novel. As such, the final aim of our realist evaluation was to produce a refined middle-range theory seeking to answer, "*what works in IOCs, for whom, under what circumstances, why, and how*?" This article was written according to the RAMESES II reporting standards for realist evaluations [29].

## Details on programmes evaluated

The 'programmes' evaluated comprised a range of IOCs as outlined in our initial realist theory paper [8]. These included arrangements of relatively low integration (i.e., buddying) through to highly integrative types such as mergers. Our interviewees had direct experience of four different types of IOC and comprised five examples of hospital groups, two alliances, three integrated care systems, and two mergers. These IOCs reflect a range of drivers and have inherent contextual differences. Although mergers result in the formation of a singular organisation, we consider mergers to be collaborative entities during the merger process until the fully merged organisation begins operating [15]. Please see Table 1 for full details on the types of IOCs and regulatory organisations included in this evaluation.

**Data collection methods.**   The realist evaluation drew on interviews conducted with an "issue network" [32] of NHS Leaders, regulators/policymakers, and frontline staff. We define an issue network as "*a broad collection of individuals possessing knowledge about the issue in question with some influence on policy outcomes*" [33].

**Recruitment process and sampling strategy.**   Participants were identified through contacts via our study advisory group and from direct contact with potential individual and organisations identified through scoping work. Participants were chosen based on their likelihood of being able to provide rich information about various aspects of the programme theory from either being engaged in implementing such arrangements themselves or delivering the policy and regulatory agendas. Participants and organisations were approached to participate via email.

*Sample*. The final sample comprised 32 interviews with 29 participants and one focus group with eight participant representatives. These interviews and focus group were conducted across England between January 2020 and May 2021. Interviews were conducted with Executives or senior staff at various IOC Programmes (n = 19), and workers at regulators and professional bodies (n = 10). In total, 17 of our interviewees had direct experience of a particular IOC while 12 participants had witnessed or were involved with a range of IOCs as part of their role. Twelve of the interviewees were already known to the study team and acted as starting points for 'snowballing'. Where case study IOCs were cross-sector in nature, the majority of these interviewees were on the acute care side of the arrangement.

The focus group was attended by eight patient representatives recruited via the Healthwatch network and were based in the Midlands, England. Their experiences covered different hospital trusts and they were involved in services to different degrees either as 'just' a patient; as someone who was an active part of the resulting hospital trust as a patient representative; or as a community representative or a volunteer with a community-based service.

**Ethical approval and consent to participate.**   The study received ethical approval from the University of Birmingham Ethics Board, as well as Health Research Authority (14[th] of

**Table 1. Participant and interview/focus group characteristics and case studies.**

| Organisational case study interviews | |
| --- | --- |
| Case study organisation and sectors | Role (Interview code) |
| Hospital Group 1 (South)—Acute care. | Director (2) x 2 |
| Hospital Group 2 (South)—Acute care. | Director (3) x 2 |
| Hospital Group 3 (South)—Acute care. | CEO (18) |
| Hospital Group 4 (South)—Acute care. | Lead (29) |
| Alliance 1 (North)—Comprising acute and community services. | Executive Nurse (10) |
| | Former CEO (12) |
| | Director (20) |
| | CEO (22) |
| | Medical Director (23) |
| | Workforce Director (26) |
| Alliance 2 (North)—Comprising acute care. | CEO (17) |
| | Director (19) |
| ICS 1 (North)—Comprising acute, social, and community care. | CEO (13) |
| ICS 2 (South)—Comprising acute, social, and community care. | Lead (14) |
| ICS 3 (South)—Comprising acute, social, and community care. | Lead (25) |
| Integrated Care Provider (ICP) (North)—comprising a formal alliance of commissioners and local providers for all of out-of-hospital services, all of community, including general practice, mental health and learning disabilities and autism services, enhanced primary care, and intermediate care. | Manager (16) |
| Merger (South)—Acute care. | Director (21) |
| Stakeholder interviews | |
| Academic & Non-Exec (1) | |
| Provider Policy Lead (4) | |
| Provider Policy Inspectorate Lead (5) | |
| NHS Provider Association (6 (x2) and 11) | |
| Professional Regulator (7) | |
| Regional Inspectorate Lead (8) | |
| Policy Transformation Lead (9) | |
| Patient Representative Lead (15) | |
| Third Sector Representative (24) | |
| Local Government Representative (28) | |
| Private Sector Representative (27) | |

January 2020). All participants provided their written informed consent to take part in the study and their identities and organisations have been anonymised.

**Interviews and setting.** Semi-structured interviews were conducted by experienced qualitative interviewers (JA, R Millar, AMR and DF). The interviews drew on realist interview methodology, with the focus on confirming, falsifying, and refining theory [34]. Questions were posed that would both directly and indirectly work towards these aims. Earlier interviews were more focused on theory gleaning and were conducted alongside construction of the programme theory in our earlier realist review [15]. Later interviews incorporated direct questions in which the programme theory was explained to the participant and they were asked about particular elements, with the intention of refining and consolidating theory directly in a manner in line with the teacher-leaner cycle [34]. This strategy was used particularly with the

participants with whom we conducted multiple interviews as well as later in the project once our theories were more developed. As we begun the data analysis while interviews were still ongoing, this allowed us to direct the interviews to further refine the theory in particular areas that were relatively unexplored by the interviews and existing theory up to that point [35]. The interview guide is available in S1 File, but was refined and developed as our interviews progressed. Due to the impact of COVID-19 shortly after the start of the research, interviews were conducted virtually over Zoom (https://zoom.us; Zoom Video Communications, Inc) or Microsoft Teams (http://microsoftteams.com; Microsoft Corporation) and recorded on an external, dedicated, encrypted audio recorder. Interviews lasted between 30–90 minutes but were typically closer to 60 minutes in length. Files were sent for verbatim transcription at a third-party transcription service.

**Data analysis.** Analysis was performed in NVivo 12 software (QSR International) by one coder (JA). Coding logic was independently verified by a second coder (RM). Coding was performed retroductively [36]. This meant that coding occurred in a deductive manner for features relating to our existing realist theory, however, for better sorting and understanding of themes, sub-codes within higher-order codes were created inductively where commonalities were identified [37]. The coding framework is available in S2 File. Transcripts themselves were not anonymised, as the coders were also the interviewers, rather, any information that would identify participants was withheld from this final article.

## Findings

### Refining the CMOCs of collaborative functioning

Our coding framework was arranged within two highest-order codes of collaborative functioning mechanisms and collaborative performance mechanisms (Fig 3). Within collaborative functioning were the following mechanisms: collaborative vs. competitive behaviour, trust and initial trust, faith and perception of task complexity, risk tolerance, perception of progress, perceived legitimacy of collaboration, conflict, conflict resolution and accountability, confidence, cultural integration, interpersonal communication, and clarity and sharedness of vision. These were taken from the prior version of the theory [3]. We also identified more detailed evidence regarding leadership qualities that affected collaborating, working either for or against collaborative efforts. These were included as contextual factors. We inductively coded new CMOCs to

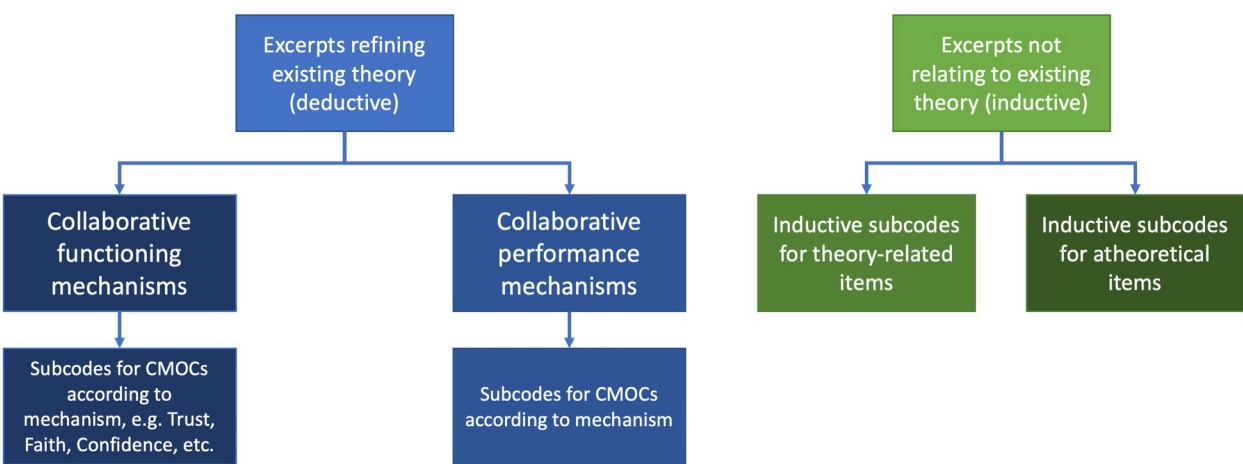

**Fig 3. Depiction of the coding framework used in the study.** This paper is focused on collaborative functioning mechanisms.

a separate code as they were identified for later reference. Novel CMOCs will be discussed in more depth within later sections that discuss the relevant mechanisms. Collaborative performance mechanisms will be explored in another paper due to the already complex and lengthy nature of this analysis.

Our interview findings identified a range of new and existing mechanisms related to collaborative functioning, what degree of evidence supported them, and any key refinements made (Table 2). Due to the number of CMOCs identified, Table 2 focuses on the mechanisms rather than the full list of CMOCs.

The following sections refine CMOCs according to the mechanism which they contain. All mechanisms depicted in Table 2 are discussed, except for collaborative vs. competitive behaviour, which tends to be an outcome for many of these mechanisms. This is because proper collaborative functioning results in collaborative behaviour as an output (Fig 2).

**Trust and initial trust.** Trust was frequently mentioned as essential to maintaining and building collaborative efforts. Particularly damaging to trust was the occurrence of conflict. In some cases, interviewees did explicitly state that conflict drove organisations to move away from collaboration and back towards competing, having "*things being done behind closed doors again*". This supports the notion that negative impacts to trust (resulting from conflict) can shift actors' mindsets back towards organisational individualism rather than collaborative thinking. When asked about the key components underlying collaboration, the following interviewee outlined how "demonstrable trust" as well as "energy" (or what we call 'faith') are essential:

> "*Your trust, you know, demonstrable trust. We've got some energy in particular areas and we've got people who are going to be able to tell a story. And we know what kind of story we want them to be telling in six months or a year's time. These are the things that are key [to collaboration]*"

(03; Director of Improvement; Hospital Group 2).

Our interviews also covered other key contextual factors affecting the mechanism of trust and these broadly matched the contextual factors that had formed CMOCs with trust in the prior version of the theory. These included the history of prior collaboration, degree of mutual understanding, power dynamic, organisational reputation, organisational sovereignty, whether staff are shared across organisational boundaries, and how collaborative success can lead to further fostering of trust in a cyclical manner. For example, one interviewee noted how the history of competition in their local health system undermined ability to build trust:

> "*There was a history of no collaboration. So sorry, it was arbitrarily. . . . Well, I've never known a place like it. Advisors just did not come together or speak well of each other. The commissioners were in a state of internecine warfare.*"

(25; Leader; ICS 3)

As for organisational reputation, there was evidence that when one partner assumed (whether warranted or not) a position of superiority and did not share resources, this decreased leaders' initial trust and desire to enter partnerships with particular organisations:

> "*. . .they'd seen how [NHS Trust] had behaved in terms of, as they saw it, the Trust that sucked up all the money, that kind of behaved in a way that was rather overbearing and arrogant as the teaching hospital, without, in their view, a lot to be overbearing and arrogant about.*"

(17; CEO; Alliance 2)

**Table 2. Description of mechanisms underlying collaborative functioning, whether they are novel or not, their degree of evidence, what refinements were made, and a supporting quotation.**

| Mechanism | Definition | Novelty/affirmation relative to existing middle-range theory and degree of evidence* | Key refinements (for aspects present in prior theory and if made) | Illustrative quotation |
|---|---|---|---|---|
| Collaborative vs. competitive behaviour | A move from competitive organisational behaviours to collaborative ones | Affirmed—Moderate level of evidence in interviews | Interviewees mentioned simultaneous drivers to compete and collaborate at any given time | *"It's felt, I suppose, being actively involved as if we're meant to be collaborating with each other and competing with each other at the same time."* (01; Academic & Hospital Group Director; Range) |
| Trust and initial trust | *"A psychological state comprising the intention to accept vulnerability based upon positive expectations of the intentions or behaviour of another"* [38]. Initial trust is trust that is manifested as a result of pre-existing contextual factors | Affirmed—Significant evidence in interviews | A perception of loss of organisational sovereignty as a context reduced trust as a mechanism, forming a new CMOC. Mistrust of regulators also formed a new CMOC. | *"So, trust is hugely at the heart of this because you have to trust that, you know, once you've agreed what you're doing that people will do what they've said they're going to do"* (26; Head of Workforce; Alliance 1) |
| Interpersonal communication | The communication and sharing of information, which supports relationship-building | Affirmed—Significant evidence in interviews | n/a | *"So it's a whole lot of things that you learn. Where it doesn't work the best, or it fails, is down to communication, understanding and that lack of awareness."* (24; Director of Care Quality—Charity Sector; Range) |
| Risk tolerance | How much risk an organisation is willing to take on with a collaborator; an organisation must be willing to engage in behaviour that could be taken advantage of by their partner | Affirmed—Moderate evidence in interviews | n/a | *"Because I think one of the things that all providers will find quite hard that they will have to do is sometimes they'll have to let go of things. That, probably, isn't necessarily what they would perceive as the best thing for their organisation. Yeah, but it probably is the best thing for patients across [Mid-UK area] in the round. And I think all organisations will probably have to give a little on some things."* (26; Head of Workforce; Alliance 1) |
| Faith and initial faith | A belief in the collaborative endeavour as a positive one and associated motivation to make it work. Initial faith refers to how much faith in is place at the start of an IOC due to pre-existing contextual factors—this is typically largely determined by the perception of task complexity and capacity, which tends to set expectations for the undertaking. | Affirmed—Significant evidence in interviews | A high perception of task complexity was found to reduce faith, in line with our prior theory. It was also found that lower capacity also lowered faith. Consistency of leadership formed a new CMOC with faith. | *"By the end of the NHS England Vanguard there was a lot of energy and passion and I remember sitting in the meetings where it was agreed that we were going to go forward and we were going to co-fund it"* (03; Director of improvement; Hospital Group 2) |
| Perception of capacity | Whether actors perceive there to be capacity to implement the collaboration | Novel—Significant evidence in interviews | n/a | *"That kind of collaborative work takes resources, you want clarity for staff as well as you possibly can, how many can you sustain in order to be effective."* (03; Director of Improvement; Hospital Group 2) |

*(Continued)*

**Table 2.** (Continued)

| Mechanism | Definition | Novelty/affirmation relative to existing middle-range theory and degree of evidence* | Key refinements (for aspects present in prior theory and if made) | Illustrative quotation |
|---|---|---|---|---|
| Perception of task complexity | How difficult actors perceive the collaborative endeavour will be to achieve | Affirmed—Moderate evidence in interviews | Local population characteristics were evidenced in the interviews to affect the perception of complexity of implementation, forming a new CMOC. The perception of task complexity was also shown to potentially reduce ability to communicate between partners, affecting collaboration and forming a novel CMOC. An unfavourable regulatory environment was found to increase perception of complexity, supporting our prior theory. | *"And that's why it's become so complex, that we're disaggregating, we're not just acquiring an organisation, we're disaggregating it whilst acquiring it."* (10; Exec nurse; Alliance 1) *"I think part of the problems sometimes mergers in particular suffer from it's almost they're sort of too big and theoretical to achieve anything"* (04; Provider Policy Leader; Range) |
| Confidence | A belief that a collaborator will behave collaboratively due to contractual or other obligation | Affirmed—Moderate evidence in interviews | There was evidence for contract enabling smoother collaboration, and a link to accountability, but little elaboration on *how*. | *"The way I look at it is it gives people some confidence that there's a structure and something you can revert back to if it all gets really difficult, but actually in practice we never really refer to the MOU and things"* (19; Director; Alliance 2) *"But you know, just using our name to get their contracts and then dropping us, yeah, it's about how you do that due diligence and build your safety nets in, and have your, you know, your memorandums of agreement for working."* (24; Director of Care Quality—Charity Sector; Range) |
| Clarity and sharedness of vision | How well-defined and to what extent the vision between partners is agreed-upon | Affirmed—Moderate evidence in interviews | A clear vision was found to increase faith, whereas sharedness of vision was shown to increase trust, which supports our prior theory. | *"'Why are we doing what we're doing?' is really key when you're setting up any type of collaboration or doing a piece of work. Being able to articulate why you really want to do, and why that solution will deliver what you're trying to achieve, is really key. That can be challenging and hard, so that's where the faith comes in, I think"* (16; Senior Commissioning Manager; Merger 1) *"... you've kind of got to have that level of trust between partners. That is everybody's areas have got the same things in mind in terms of endgame."* (23; Medical Director; Alliance 1) |
| Perception of progress | Whether actors perceive they are advancing towards achievement of the goals of the collaboration | Affirmed—Moderate evidence in interviews | Having ongoing evaluation was essential to perception of progress, as well as implementing 'quick wins' early on to drive trust. A stronger perception of progress was also shown to increase faith. A novel CMOC was identified whereby a context of peer pressure can affect perception of progress and thereby faith. | *"And if you spread your resources too thinly you're trying to do too many things and none of them quite come to fruition you just get into this, 'Well, it's never delivered anything has it so why are we bothering?'"* (19; Director; Alliance 2) Novel CMOC: *"And there's an element of peer pressure; you don't want your programme to be the one that's falling behind, because you'll look bad in front of your colleagues, so of course there's some of that."* (17; CEO; Alliance 2) |

*(Continued)*

**Table 2.** (Continued)

| Mechanism | Definition | Novelty/affirmation relative to existing middle-range theory and degree of evidence* | Key refinements (for aspects present in prior theory and if made) | Illustrative quotation |
|---|---|---|---|---|
| Cultural integration | How well actors between organisations are aligning in terms of attitudes and behaviours | Affirmed—Moderate evidence in interviews | A better cultural integration process, such as performing cultural due diligence prior to beginning the IOC, improved performance of the collaboration, as found in our prior literature-based theory. Cross-sector working was shown to increase difficulty of cultural integration, in line with our prior theory. | "*So all that kind of cultural due diligence and understanding how it works and so on, you know, things happen before the decisions have merged, which probably then gave us a head start once the merger started and then the head start afterwards.*"(01; Non-Exec Director; Range) |
| Conflict | The perception by organisational actors that they are in opposition to collaborators in some way | Affirmed—Moderate evidence in interviews | Conflict negatively impacted both trust and faith in the interviews, which is in line with findings from the prior theory. | Link to faith:<br>"*I was part of and experienced an awful lot of minor power politics between organisations, which I just found frustrating and served no purpose, but just sapped energy.*" (12; CEO; Alliance 1)<br>Link to trust:<br>"*A good example I suppose is when we went out to market engagement and public engagement on whether we should have an MCP [Multispecialty Community Provider]. They drove a huge deal of conflict in the system just doing the consultation. The local authority were particularly not bought in, they lost faith completely then, and trust.*" (16; Senior Commissioning Manager; Merger 1) |
| Conflict resolution and accountability | Processes and attitudes in place that can lessen the severity of conflict | Affirmed—Limited evidence | Accountability was found to be impacted by leadership approach and a new CMOC was identified in which unaccountable leaders reduced the ability to resolve inter-organisational conflict. | "...*if there's one organisation that's so adamant that what you're proposing is a bad idea and then prepared to hold out against it for us ultimately if that was the case there's a dispute resolution mechanism we could go through and ultimately the end point of that dispute resolution mechanism is deciding not to be part of [the alliance] anymore.*" (19; Director; Alliance 2) |
| Perceived legitimacy of collaboration | How actors perceive the collaboration as an authentic means for achieving improvement | Affirmed—Moderate evidence in interviews | Perception of legitimacy was improved by a context of stakeholder involvement, which supports our prior theory. It also encompassed the concept of mandated collaboration being, in many cases, perceived as a takeover. This mechanism impacts faith. | "*So, I guess our first collaboration with another organisation that was in some difficulty was probably not seen as collaboration and was seen as a sort of takeover of another executive team*" (20; Director; Alliance 1) |

*Limited evidence means 1 or 2 codes were present in interviews, moderate means 3 or more codes, and significant means 5 or greater codes.

These factors concurred and supported our existing CMOCs. Only organisational sovereignty was not represented with a CMOC in our prior theory. Our novel CMOC relating to trust suggests that:

*A feeling of loss of organisational sovereignty (context) → reduced trust (mechanism) → reduced collaborative behaviour (outcome)*

Power was also elucidated further than in our prior analyses. Power was framed as always intrinsic to collaborations, and that they were rarely truly ever equal partnerships:

*"It's also often said that partnerships are a relationship of equals and I'm not sure they are to be fair. They're just because, you know, organisations are bigger than others."*

(01; Non-Exec Director; Range)

Interviewees recorded that they were able to achieve more equal status by demonstrating their usefulness as a partner and by accomplishing tasks together:

*". . .now we've been seen more as an equal in the system but we bring added value and we can, with people's agreement, we can still link them in to some of the other things that we provide*

(15; Patient Representative Lead; Range)

A number of leadership behaviours also impacted trust beneficially. These included empathetic leadership, listening and reacting appropriately, visibility, negotiation and diplomacy, showing vulnerability, and persuasiveness. These could form a CMOC whereby:

*Beneficial leadership behaviours (empathetic leadership, listening and reacting, showing vulnerability, persuasiveness) (context) → improved trust (mechanism) → greater collaborative behaviour (outcome)*

For example, for persuasion, several leaders emphasised how important it was to be able to be persuasive when it came to building a shared vision and for people to engage with the collaboration:

*"A lot of it is a bit of shuttle diplomacy in a way; you sort of go between people persuading these people to move this way and them to move a bit that way so you get them all into a place where they're close enough together that they can work together."*

(19; Director; Alliance 2)

Likewise, for empathetic leadership, interviewees reiterated the importance of enabling people to be supported to enter into that collaborative mindset:

*". . . it is about your kind of, you know, having that personality, the emotional intelligence, just the skills around knowing when someone's unhappy, being able to sort of understand and bring people along and spending time with people. It's not just transactional, you've got to spend a lot of time on the sort of OD [organisational development] side of things with something like this"*

(17; CEO; Alliance 2)

**Interpersonal communication.** Interpersonal communication, and the power of informal interactions that build genuine relationships, often taking the form of face to face meetings to find common ground, was found by interviewees to be the key mechanism that leads to improvements in trust, supporting our existing links between CMOCs [3]. One NHS Leader reflected this in the following quote:

*"So going to meet with them and understanding what was happening in [Hospital 1] and [Locale 1] and [Locale 2] and so on, and then, suggesting, "look, we've all got a lot in common, why don't we get together?" and so we did."*

(17; CEO; Alliance 2)

Our prior hypothesis that interpersonal communication would be negatively affected by larger geographical distance was also reflected in the interviews, whereby greater distance could make it harder to build trust, due to difficulties in having regular interpersonal communication. One interviewee reflected that:

*"It's never going to end up in a kind of merger or acquisition because we're too far away"*

(03; Director of Improvement; Hospital Group 2).

**Risk tolerance.**   In this theory, risk tolerance connects trust and confidence to the output of collaborative behaviour, by enabling the organisations to engage in 'risky' behaviours which could otherwise be taken advantage of by a less trustworthy partner in a competitive environment. Interviewees concurred that being willing to take a risk supported collaborative behaviour. Intrinsic to the nature of taking risk was the understanding that one organisation had to act in a vulnerable manner (i.e., 'give something up') with the expectation of potentially receiving something in return from the collaboration later:

*"Because I think one of the things that all providers will find quite hard that they will have to do is sometimes they'll have to let go of things. That, probably, isn't necessarily what they would perceive as the best thing for their organisation."*

(26; Head of Workforce; Alliance 1)

Another leader highlighted the mental process their organisational board went through when establishing what was risky (or not) to give away in the collaboration, and how a patient focus was important for rationalising those decisions:

"*So, I say the true test for me of how collaborative we want to do is something that's really important to us and we're willing to give it away because we think that there is a greater good for the patient and the population in giving it away. We usually find some data that tells us we're totally brilliant at it and nobody else could be more brilliant at it than we are. But the true test is, 'Are you willing to give something away?'"*

(20; Director; Alliance 1).

As such, it is clear that building risk tolerance to be able to 'give something up' is essential to changing mindset from competitive to collaborative thinking. Key to the theory is also that building trust also increases risk tolerance. One interviewee, when asked directly about the theory in line with the teacher-learner cycle, did think that trust could help build risk tolerance, stating that:

*"So yes, completely agree with that. In this context around kind of, okay, so you build the trust, you're willing to engage in that kind of risk"*

(03; Director of improvement; Hospital Group 2).

**Faith and initial faith.**   The concept of faith featured prominently in our interviews. Some interviewees referred to faith as an 'energy' or 'zest' for collaboration, while others referred to faith as 'seeing value' in collaboration. In many cases, faith served to reduce anxieties about

task complexity in line with our prior CMOCs. One interviewee said: *"if there's energy there, I'm less concerned about the complexity of what people need to do because they'll work it out"* and they went on to emphasise the key ingredients for collaboration: *"So you've got trust, you've got energy."* (21; Director of Clinical Service; Merger). This suggests that having faith in a collaboration means that those involved would be willing and able to work through any complexities that emerged.

Another aspect confirming our prior CMOCs, was the notion that overpromising (stating that something could be accomplished that is not realistically achievable) could reduce faith, and that staff turnover could also have a negative impact. For example, the following leader illustrated the impact of overambition on faith:

*". . . that's the sort of thing which is too big, too contentious, is going to take you too long and you're going to be five or six years into this not really feeling like it's actually delivered anything and that's when people are going to go, 'Why are we. . .? We've not seen any outcome from this so what are we doing?'"*

(19; Director; Alliance 2)

Another key CMOC in our prior iterations of the theory was that the consistency of leadership approach can improve faith. This was reflected in the following quote: *"I think if you've got a bit of stability but it is consistency of message, of priorities, you know, consistency of approach really is what I'm really talking about there"* (18; CEO; Hospital Group 3).

As such, the following CMOC can be formed:

*Consistency of leadership (context) → improved faith (mechanism) → greater collaborative behaviour (outcome)*

Leadership behaviours were also identified in the interviews as increasing faith. These included fostering talent, being visible, espousing local benefit of collaboration, persuasiveness, commitment, consistency of approach, removing noncooperative actors, listening and reacting appropriately, and reframing. For example, the following quotation supported 'listening and reacting appropriately':

*". . .if that [leader] who's coming in doesn't listen to what's good in the other organisation as well as what needs to be put right and doesn't recognise kind of, you know, what's good and needs to be put right in their own organisation, again that's not going to work, because people will resent it"*

(07; Nursing/Midwifery Regulator; Range).

These behaviours could all form part of the context in CMOCs that improve faith, thereby increasing collaborative behaviour. Negative leadership behaviours and attitudes included being missing in action, having a lack of accountability and being overly resistant to change. These could reduce faith in the same way as the aforementioned positive behaviours. Resistance to change was exemplified by the following quote, whereby change in leadership to a more resistant leader, in a context of others also already being resistant, undermined a collaborative effort:

*". . .the moment those noises [of resistance] became louder was when it became harder to progress on a number of fronts because there was significant resistance from a number of different parties at [NHS Trust] despite their Chief Exec at the time being very supportive. They then*

*had a change of Chief Exec. The new Chief Exec came in with support from the [local] political establishment—I'll just put it that way."*

(03; Director of Improvement; Hospital Group 2).

Lack of accountability by leaders will be discussed in the conflict mechanism section.

Interviewees viewed the regulatory environment as also important in determining system-wide faith around collaboration. In some cases, it was reported that providers were promised funds from central bodies to deliver certain collaborative arrangements, funds which later failed to materialise. This led to conflict, resentment, and created mistrust vertically in the system. One Leader stated *"in the end, we weren't supported financially in the way that I was led to believe we were going to be. And so, we had a choice about whether we stayed doing what we were doing or withdrew"* (12; Former CEO; Alliance 1). As such, such broken promises clearly have a negative impact on faith. Mistrust in this example focuses more on trust between providers and regulators, rather than between partners, as is the case with other aspects (e.g., organisational culture) discussed here.

*Mistrust of regulators (context) → reduced initial trust (mechanism) → increased conflict (outcome)*

Mistrust of regulators is likely to also lower faith in collaboration as opposed to affecting the trust between partners. As conflict affects both trust and faith, it is clear how the above CMOC can tie into loss of faith further down the chain:

*Increased conflict (context) → reduced faith (mechanism) → reduced collaborative behaviour (outcome)*

*Perception of task complexity*. Task complexity was also raised in the interviews. In line with our prior CMOCs, the number and size of organisations involved in the collaboration were seen as increasing perception of task complexity. For example, one interviewee stated that they were "in a five/six-way partnership [. . .] I suspect it will emerge as one of the most complex transactions that anybody's worked on" (10; Exec Nurse; Alliance 1). This interviewee reflected that the amount of work it took to coordinate across that many partners made it extremely difficult. Another reason put forward by our interviewees why larger collaborations lead to increased perception of task complexity is that larger collaborations undermine the ability of those involved to develop meaningful relationships with those they needed to collaborate with. For example, one interviewee reported: "I don't think it will be possible for me to do my job in the way that I like to do it if I was responsible for anything bigger than I am now, because I just don't think you can have that relationship that I've got with the staff" (17; CEO; Alliance 2). This results in the following CMOC:

*Larger organisational size or more involved partners (context) → increased perception of task complexity (mechanism) → reduced interpersonal communication (outcome)*

This means that complexity may also impact on trust, further down the causal chain.

The impacts of regulation were underdeveloped in the prior stages of our theory, and, as such, we sought to focus on this aspect in the interviews. In our prior theory, regulatory environment and its complexities were found to both increase and decrease perception of task complexity, and thereby faith. A highly unfavourable environment was posited to make key actors, e.g., leaders, perceive it as excessively difficult to collaborate. This was also the case in the interviews, which focused on the English policy context, where there are conflicting incentives to collaborate in some cases, and compete in others [39]. This made one Leader reflect that *"there's also a risk of setting partnerships up to fail"* (01; Non-Exec Director; Range).

*Regulatory environment unfavourable to collaboration (context) → increased perception of task complexity (mechanism) → decreased initial faith (outcome)*

With the interviews having taken place in England, leader interviewees had heard of the rather public blocking of other collaborative efforts by the England's Competition and Markets Authority (CMA), such as the blocking of the merger of Bournemouth and Poole hospitals. This lowered their initial faith sufficiently to disincentivise considering any such arrangements themselves. Restrictions imposed by the CMA on certain types of collaboration, usually between providers of the same services, made actors seek forms of vertical, rather than horizontal, integration and collaboration. As a result of this policy, in some cases, collaboration between providers of the same services seemed worth actively avoiding as the requirement for 'faith' was likely too high. Interviewees reported meeting with the CMA to ensure that types of collaboration they were considering were unlikely to be blocked. However, the health system was seen as shifting overall towards collaboration: "*I think people do see collaboration or partnership working as a means to solve their problems, you know, problems that they can't resolve on their own*" (06; NHS Providers; Range).

A new CMOC was also developed for the perception of task complexity, relating to local population characteristics as a context for collaboration. Leaders noted that:

> "*I've also got an affluent educated population which means then I can then focus on my pockets rather than having to bottom up [. . .] I'm playing the cards I've got but they probably are quite a unique set which is why the solution we've got probably is quite unique and may not work elsewhere.*"

(14; Leader; ICS 2)

This leader indicated that having this demographic makeup in their locale made it easier to implement their collaboration as it reduced the health pressures they were facing. As such, a:
*More educated or affluent local population (context) → reduced task complexity (mechanism) → improved initial faith (outcome)*

*Perception of capacity*. Faith was also found to be impacted negatively or positively by perceptions of capacity, with interviewees stating that they were not motivated to collaborate in cases where there was not the capacity to do so. Having 'capacity' was often expressed in terms of financial resources. For example, one such reflection was that "we have some financial challenges, and they knew that nobody would be beating the drum to say, 'Well why don't you merge?'" (03; Director of Improvement; Hospital Group 2). A similar sentiment was echoed by another interviewee: "So, I think once you started to remove the money barrier [. . .] they will say, 'What is it that you need and how can I make that happen?' So, there's I guess quite a lot for me about. . . I think organisations are quite good at collaborating at a place-based level" (20; Director; Alliance 1).

Likewise, another highlighted that capacity concerns can occur later in the lifecycle of a collaboration: "*they've had that capacity taken away because they had their budget cut*" (11; Policy executive; Range). In these examples, resource availability (whether temporal or financial) for collaboration forms part of the context which shapes whether key actors perceive there to be capacity to engage in collaboration, thereby improving initial faith and faith later in the process. Another interviewee succinctly spelled out this conclusion: "*you can't just cut [funding] and expect people to change and have the resilience and energy, without any input or support or development*" (24; Director of care quality; Range). This results in the following CMOC:
*Enhanced resource (e.g., funding, regulatory support) availability (context) → improved perception of capacity (mechanism) → improved faith (outcome)*

**Confidence.**   Formalisation and the fostering of collaborative behaviour through contracts was highlighted in the interviews as an enshrinement of 'principles' or 'ways of working' in

contract. Memoranda of understanding (MoU) were typically used in arrangements outlined in our interviews. These lighter-touch contracts were put forward as supporting rather than undermining collaborative working. One interviewee stated that:

> "*There's some practical stuff in there that we use regularly which is helpful, so in terms of 'how do you make decisions and approve business cases' and things like that, which is set out in the MOU and we use that regularly. But because it's a useful way to go about it actually I think that sort of governance bit is important at the start to give people that confidence*"

(19; Director; Alliance 2).

This statement indicates that much of the governance structure upon which collaborative behaviour depends can be enshrined in contract. Contracts also assisted partners in enforcing accountability in many cases, understanding *who* was responsible for *what*. This interacts with the risk tolerance mechanism by enabling a reduction in risk of engaging in collaborative behaviour. For example, one interviewee noted that they made their partner "*rewrite their MOU and be very crystal clear where a delegated authority is given*" (13; CEO; ICS 1) due to a joint committee structure that "*have never made a decision ever*". Overall, there was moderate support for our prior CMOCs regarding confidence, formalisation, and confidence's connection to risk tolerance.

**Clarity and sharedness of vision.**   How clear the vision is, as well as how shared it was across the collaboration, were connected in the interviews to different mechanisms. In line with our prior refined theory, clarity of vision served to increase faith by minimising miscommunications, whereas the degree to which the vision was shared across collaborators was seen to increase trust by avoiding likelihood of conflict. For the former, one interviewee stated: "*Being able to articulate why you really want to do, and why that solution will deliver what you're trying to achieve, is really key. That can be challenging and hard, so that's where the faith comes in*" (16; Senior Commissioning Manager; ICP). As for the relationship between sharedness of vision and trust, another interviewee stated "*. . . the thing you [need] is aligned purpose. There actually wasn't a sense of aligned purpose. That then percolated into trust. And then that affects a number of other things*" (03; Director of Improvement; Hospital Group 2). This ratifies our prior theoretical assumptions.

**Perception of progress.**   In our refined theory, feeling like progress was being made increased faith by serving as a motivator to continue with the collaboration. If people perceive the IOC as making progress, they will want to keep working on it, and vice versa. Overambition was found to also lessen the perception of progress, as people would be less likely to achieve overambitious aims. Likewise, evaluation was frequently raised as being essential to keeping track of progress. One interviewee stated: "*an organisation or individual, an organism, doesn't improve without knowing that it's doing the right thing. And getting feedback and hopefully having that being positive reinforcement is hugely important.*" (12; Former CEO; Alliance 1). As discussed in our prior theory, the notion of 'quick wins' early on in the collaboration was also brought up, with an interviewee mentioning that implementers should "*get a couple of teams working together on a specific problem, like, 'how do you redesign this service', and suddenly everyone sees the point of collaboration and what it really means; so I guess focusing on the practical and sort of some quick wins*" (04; Provider Policy Leader; Range). This, together with this excerpt from another practitioner: "*if people don't see things getting any different then it just becomes a talking shop and people start drifting away*" (02; Director; Hospital Group 1) also draws a connection between faith and the perception of progress, supporting our theory.

Another new CMOC related to a perception of peer pressure between collaborators, which can drive 'perception of progress' and affect faith. This was reflected in the following quotation: "*you don't want your programme to be the one that's falling behind, because you'll look bad in front of your colleagues*" (17; CEO; Alliance 2). This seems to have the potential to both reduce and increase faith depending on whether actors perceive they are progressing well. As such:

*Perception of peer pressure (context) → renewed focus on perception of progress (mechanism) → increased faith if progressing well or decreased faith if progressing poorly (outcome)*

**Cultural integration.**    Culture was seen by interviewees as a collection of behaviours and attitudes shared within organisations. These attitudes and behaviours could be conducive or unfavourable to collaboration and to change. Culture was viewed as essential to collaborative success, at a similar level of importance as having the right leadership. The importance of cultural integration was reflected in the following quotation: "*A single board trying to run what, in essence, was four hospitals with four cultures from a single base, wasn't going to work*" (10; Exec Nurse; Alliance 1). Having a larger degree of difference in culture across organisations made it more difficult to collaborate.

One of our prior CMOCs posited that cross-sector collaborations would involve more people from different professional backgrounds working together and hence would have more difficult cultural integration due to differing work cultures. This was also reinforced by interview findings, which supported the notion that working together across services was more difficult than working within them. This example highlights this perception and perhaps also hints at some of the power dynamic at play between different healthcare sector organisations that collaborate together: "*there's such a massive skills and knowledge deficit between people who run hospitals and people who work in systems and work in primary care. In these big Trusts, like ours, I think it's even more challenging to get that.*" (14; Leader; ICS 2).

As for sharing a culture through an IOC, a "culture of improvement" was seen as essential in higher-performing organisations. Often, sharing such an 'improvement culture' was often the impetus for such collaborations in the first place, either through regulatory mandate or voluntarily through a desire to improve performance in the local health system. One leader put forward that:

"...*if you want to have an organisation where the culture is of improvement, people have to, I think, really understand that you're not just picking up the good things, you know, you're saying what the difficulties are, and you're reporting if you've not met what you set out to achieve.*"

(12; Former CEO; Alliance 1)

This meant that the culture of improvement is underpinned by an honest and reflective approach to problem solving and progression. Further underlying the culture of improvement is removal of 'toxic' cultures exemplified by bullying and overburdening management, as well as "*learned helplessness*", which were also raised in the interviews. In the various types of collaboration we covered, there were examples given of organisations which had managed to work together despite having different cultures, cases where cultural differences proved impossible to overcome, as well as some where cultures had been successfully changed. Essential to integrating cultures successfully was the approach by leadership; particularly key was the requisite cultural due diligence initiated by skilled leaders prior to engaging in collaboration. However, in mandated collaborations, for example, organisations may not have the opportunity to perform these preparatory actions. One leader outlined this in the following excerpt:

*"So all that kind of cultural due diligence and understanding how it works and so on, you know, things happen before the decisions have merged, which probably then gave us a head start once the merger started and then the head start afterwards. Lots of organisations don't get that preparation time so they get thrust into something."*

(01; Non-Exec Director; Range)

Outside of cultural due diligence, culture was also found to be largely formed by leadership. This dynamic was put forward in our initial realist theory paper [8] and is reflected in the following quote:

*"And [NHS Trust] had longevity of leadership for 10/11 years. But they created a culture that was fundamentally flawed, because they created a culture of telling and doing to, rather than empowering and listening and enabling people to find solutions to the problems that they were facing. So, longevity of leadership is one thing, but it has to be the right leadership, it has to set the right tone and the right culture"*

(10; Exec Nurse; Alliance 1).

It also became clear that a key performance improvement mechanism is the fostering of a productive culture. Other leadership teams can have an influence on the collaborative organisations and can help to instil a culture of greater productivity, as reflected by the following quote: *"the [Trust] has partnered with it's not a group, it has partnered with another trust locally. And the Chief Executive there is quite astonishing and she has just driven through some major changes, her own internal improvement methodology basically which she thought about in the [Trust] and then expanded into the trust that they've partnered with"* (11; Policy Exec; Range).

As such:

*Leaders fostering a productive culture (context) → improved organisational culture (mechanism) → improved collaborative behaviour (outcome)*

**Conflict.** In our prior theory, we had emphasised that conflict could occur both across and within organisations, and could affect both trust and faith depending on the type. The impact of conflict on faith is reflected in the following quotation: *"I was part of and experienced an awful lot of minor power politics between organisations, which I just found frustrating and served no purpose, but just sapped energy [. . .] so in the end, I became a bit frustrated with it all and stepped away"* (12; Former CEO; Alliance 1). This shows that instances of conflict leading to a complete loss of faith for individuals could potentially propagate to cause a loss of faith across organisations in the IOC. Conflict between organisations and its effect on trust was also pronounced, where *"the conflict then gets played outside of the room, in corridor conversations and email conversations etc."* (10; Exec Nurse; Alliance 1). Conflict was perceived as being always ongoing to some degree, due to the history of interpersonal relations within local systems and the number of interpersonal connections that must be maintained in these collaborations. However, better-functioning collaborations could minimise the impact of conflict on trust and faith by implementing better conflict resolution and accountability processes.

**Conflict resolution and accountability.** Mechanisms for resolving conflict and improving accountability were raised in the interviews. When conflict occurs, conflict resolution and accountability mechanisms are able to reduce the impact of conflict on trust in particular, but potentially faith as well. Conflict could often be resolved through informal mechanisms, such as by having a person dedicated to managing the relations in the IOC who could step in to resolve problems informally. ICS 3 had such a manager dedicated to IOC relations whom we interviewed:

"*So, when they have a spiky moment with each other, they know they're going to have to have a conversation with me together about that interaction. So therefore, and not from a culture of fear, but, I'm but like, 'I don't think you both treat each other very well. And there was an audience that watched that, that's not how we work here, what's going on?'. Only a few times, you have to do that, and quickly they go, 'I will not tolerate that behaviour, I will not tolerate speaking negatively'. And you've got to keep reminding them of that*"

(25; Leader; ICS 3)

Formal conflict resolution mechanisms were also outlined by interviewees. One interviewee noted that if a severe conflict occurred, "*there's a dispute resolution mechanism we could go through and ultimately the end point of that dispute resolution mechanism is deciding not to be part of [the collaboration] anymore*" (19; Director; Alliance 2). This supports the notion within the programme theory that conflict can lead to dissolution.

For accountability, leadership was a big component, where leadership needed to be held to account for problems during the collaborative process. For example, one interviewee stated that prior to their collaboration:

"*The number of staff suspended was massive, because the leaders blamed the staff when things went wrong, instead of saying, 'The system didn't enable the staff to operate safely.' That was the culture that was in, that was prevalent in the organisation. So, the leaders didn't take any accountability for the systems that they'd put in place; they just blamed the staff for getting it wrong*"

(10; Exec Nurse; Alliance 1).

This could be considered a new CMOC which will affect trust further down the chain:
*Unaccountable leaders (context) → reduced ability to resolve conflict (mechanism) → increased conflict (outcome)*

**Perceived legitimacy of collaboration.** The perceived legitimacy of the collaboration was added in the refinement stage of this analysis, based on case study findings. Whether a collaboration was voluntary or mandated could increase legitimacy in the case of the former and reduce it in the latter. Support by stakeholders could also increase the perception of the collaboration's legitimacy. This was reaffirmed in the interview findings, with reputation of the partnership type also being impactful to its perception of legitimacy. In England, ICSs, a form of collaboration being linked to notions of privatisation in the media, had been viewed as particularly damaging [40]. This was noted by one interviewee: "*what we then did was say okay, we have got to make this happen but at the time ICSs were pretty toxic, all about privatisation and all this nonsense about privatisation and all this rubbish*" (13; CEO; ICS 1). This resulted in the interviewee trying to delay their organisation becoming part of an ICS until such perceptions had dissipated, to avoid public backlash.

Engagement with service users was found to increase legitimacy and thereby faith. Leaders often found it helpful to have service users involved to keep the focus of the collaboration on the outcomes that really matter, as reflected in this excerpt: "*having patients in the room, having them help shape how these pathways pan out is really helpful in cutting out some things that basically don't add value*" (13; CEO; ICS 1). However, as reflected by the quote from the leader above, a major fear voiced by the patient representatives was that greater collaboration would lead to privatisation, which undermined the legitimacy of many collaborations in their eyes. As for faith, one interviewee stated that "*involving your service users, your stakeholders makes a massive difference, you know, and that can break down so many barriers. And it can allow you*

*to be more proactive, rather than reactive*" (24; Director of Care Quality—Charity Sector; Range). These findings support the existing CMOCs.

## The impact of mandating collaboration

Interviewees were largely sceptical of the notion of mandated collaboration—a current policy direction in England. A mandate has historically been used to enforce limited 'buddying' arrangements to support poorly performing organisations through sharing organisational learning [41]. Additionally, mergers have also been mandated for poorly performing organisations. Current policy is now to mandate collaboration on a large scale to establish ICSs in statutory form across England in 2022 [42].

Interviewees were generally of the opinion that *"you can't force trust and you can't force a good kind of relationship."* Another stated "*that investment in time in building relationships, building trust, is hugely important. I think the challenge the NHS has got the moment is it hasn't got that luxury. So, you know, they're going to have to do it, like as a mid-air refuel.* (26; Head of Workforce; Alliance 1)*"*. The interviewees suggest that enforcing collaboration is a recipe for conflict. In one example, a provider was asked by a regulator to 'step in' to 'collaborate' with another organisation, but in fact "*they wanted us to step in with a managed contract and actually start to run the organisation. So, it didn't start from a coming together of partners that could see a better future together*" (10; Exec Nurse; Alliance 1). This resulted in the other organisation's executive team all resigning, as it was perceived as a hostile takeover rather than any kind of collaborative effort. Another NHS leader warns that mandated collaboration increases the chance of unintended consequences, which could have a negative impact:

> "*If you're not [ready to change] you can buddy up until the cows come home, things feel forced. You can't force something on an organisation without getting an unintended consequence.*"
>
> (25; Leader; ICS 3)

In our realist review, we identified some CMOCs related to the impact of mandated collaboration, implying that a context of mandated collaboration leads to a reduction in initial trust and thereby potential conflict. As evidenced by the above quotations, these CMOCs are supported by the interview findings. These findings for mandated collaboration present implications for ICSs and Primary Care Networks, which are current policy mainstays in England.

## Discussion

### Formulation of action statements from CMOCs

The refinement of CMOCs in this final step of our realist theory allows for formulation of action statements which connect practical steps organisations can take to improvements in theoretical constructs such as trust, faith, and confidence [24]. These are outlined in Table 3 below.

### Contrasting interview with literature findings

This study has enhanced our understanding of the factors underpinning the effective functioning of IOCs, including the key mechanisms and contextual factors, and how these interact to form a chain of generative causation. Likewise, this study sheds light on which leadership attitudes and behaviours support (such as being empathetic and showing vulnerability) or constrain (e.g., through a lack of accountability) the maintenance and development of IOCs, as

**Table 3. Action statements about how to maximise collaborative functioning and thereby performance.**

| Cluster | If (context)... then (outcome)... because (mechanism)... 'action' statements |
|---|---|
| Leadership behaviours and attributes | • If beneficial leadership traits and behaviours including empathetic leadership, listening and reacting, fostering talent, visibility, showing vulnerability, espousing local benefit, learning from others, persuasiveness, commitment, consistency of approach, fostering positive culture, and reframing are in place, these can increase collaborative behaviour by enhancing trust and faith<br><br>• If negative leadership traits and behaviours such as being 'missing in action', not being held accountable, and being overly resistant to change are occurring, they can reduce incidence of collaborative behaviour because they significantly decrease faith and willingness to collaborate<br><br>• If a "culture of improvement" for organisations is fostered, such as through use of practical tools such as performing cultural due diligence, this can promote engagement in collaboration by improving faith<br><br>• If workers or other leaders who do not have faith in the vision for the IOC are removed, this can help promote collaboration later by improving shared vision and therefore faith<br><br>• If the collaboration works towards sharing improvement programmes and strategies across organisations, then organisational performance improvement can occur because it can improve collaborative behaviour |
| Enhancing trust and interpersonal ties | • If organisations work on understanding and mitigating the impact of negative prior experiences in collaborations, then they can improve initial trust because they can reduce internal organisational conflict<br><br>• If organisations can foster a mutual understanding between organisations by formulating and instilling a shared vision, then they can reduce conflict, because this creates a 'team' rather than 'us vs. them' atmosphere<br><br>• If the collaboration can deliver 'quick wins' at the beginning of the lifecycle of the collaboration, then they can improve trust and faith, because they avoid a sense of inertia and maintaining forward momentum and 'energy' within and between organisations<br><br>• If collaborations can understand and mitigate the impact of the regulatory environment, then they can improve faith, because it lowers the workforce's perception of complexity<br><br>• If the collaboration can prioritise interpersonal communication between organisational actors, face-to-face where possible, incorporating informal chats, then they can improve trust, because this helps to build genuine interpersonal relationships<br><br>• If there is significant geographical distance between partners, then this can act as a barrier to trust formation, because it can undermine ability to have informal interactions<br><br>• If IOCs can allow flexibility and a degree of autonomy within itself, then more trust may be built, because it avoids feelings of a loss of organisational autonomy (and conflict) |
| Risk tolerance | • If the collaboration implements an appropriate degree of formalisation (see formalisation below) then it can increase risk threshold of involved organisations, because the formalisation lowers the inhibition of an organisation to put itself in a vulnerable position relative to its partner |
| Faith and initial faith | • If involved organisations include service users and frontline staff in its design, then faith will be improved because this helps keep the vision clearly focused on key outcomes<br><br>• If a clear vision is maintained from the outset, and is understood by all partners, with clear outcomes and a logical path for their achievement, then this improves trust because it helps to avoid conflict<br><br>• If the collaborations keep an appropriate level of ambition, faith may improve, because realistic ambitions prevent feelings of failure when lofty goals are not achieved (which would lower perception of progress)<br><br>• If collaborations involve very large organisations, or many partners, then this can reduce faith, because it causes an increase in perception of complexity<br><br>• If organisations can ensure that there is a perception of progress (forward momentum) then this will contribute to faith being maintained because there is ongoing evaluation and implementation of achievable milestones<br><br>• If the organisations implements 'champions' in the IOC then this can help to spread the faith in the collaboration because they help share the vision within the workforce<br><br>• If collaborations can prevent a high degree of staff turnover, then they can prevent a loss of faith, because every staff member that leaves needs to re-learn the vision of the collaboration<br><br>• If organisations can understand the reputation that the chosen IOC form has in the public (e.g., in UK, whether it is associated with privatisation) then they can improve faith because they may be able to better understand the impact on the workforce |
| Managing mandated collaborations | • If those involved in mandated IOCs can acknowledge that mandated IOCs, which can arise as a result of regulators stepping in, organisational failure, and external events such as COVID-19, or takeover, usually manifest an unequal power structure, then they can reduce the negative impact of this on trust because they pre-emptively understand that additional conflicts may arise and can take steps to mitigate this<br><br>• If those implementing mandated collaborations understand that building relationships and a truly cooperative environment cannot be forced, then they may be able to guarantee a certain degree of collaborative behaviour through increased confidence, because they can use contractual mechanisms to enshrine collaborative behaviour in contract<br><br>• If a collaboration is mandated, then efforts should still be undertaken to build proper working interpersonal relationships in the longer term, because this improves trust and thereby genuine collaborative behaviour |

*(Continued)*

**Table 3.** (Continued)

| Cluster | If (context)... then (outcome)... because (mechanism)... 'action' statements |
|---|---|
| Confidence and formalisation | • If organisations can ensure that there is an appropriate level of formalisation of the IOC through contract then they can increase collaborative behaviour, because, for more complex, integrative, or mandated collaborations, greater formalisation mitigates risk between partners when engaging in collaboration. However, for smaller, less integrative, or voluntary collaborations, over-formalisation can undermine trust<br>• If collaborators can perform due diligence around potential areas of disagreement, and enshrine accountability mechanisms in contract, then this can improve trust because it can later help to amicably resolve conflicts |
| Managing and resolving conflict | • If shared and clear accountability is in place in the IOC, then conflict can be avoided because of the improved understanding of where accountability lies<br>• If the IOC has clear dispute mechanisms in place, with adjudication by an unbiased third party, then trust can be improved because conflicts are more easily resolved<br>• If there is a clear and shared vision in place, then this improves trust, because it helps to prevent conflicts occurring |
| Ensuring capacity for implementing the collaboration | • If organisations recognise that, pragmatically, implementing a collaboration requires significant time, effort, and financial input, then they will have improved faith, because their perception of progress will not be negatively impacted<br>• If key IOC actors (i.e., leaders, architects of the collaboration, senior staff) understand that initial performance drops may occur while resources are redirected towards the implementation of the collaboration, then collaborative behaviour will be enhanced because losses of faith will be less severe<br>• If organisations begin a collaboration without first ensuring there is adequate capacity, then this can undermine faith and trust between partners, because their perception of progress will be negatively impacted. Additionally, if funding to implement collaborations is sourced externally and it is not forthcoming, then faith and trust can be critically affected because capacity is no longer sufficient |
| Cultural integration | • If a shared culture can be fostered between organisations then trust can be improved because it helps to avoid culture-related conflict<br>• If the IOC is cross-sector or cross-service, then it may have a greater cultural divide because of differing professional backgrounds, which makes cultural integration more difficult<br>• If IOCs perform cultural due diligence prior to implementing the collaboration and implement a plan for cultural integration, then they can improve trust because it avoids unforeseen culture-related conflicts occurring later in the process |

well as the impact of regulation on the functioning of IOCs. These were able to bridge gaps in understanding in our prior theory [3, 15]. As a result, the majority of the CMOCs and their underlying mechanisms were affirmed and no outright refutations were identified (Table 2), strengthening the findings of our prior review. Several new CMOCs were formulated and outlined throughout this article, which interacted with the existing mechanisms. This study also made clear just how complex and interconnected many of these mechanisms are—with culture and leadership, while both being aspects of *how* collaborations function, also becoming linked into collaborative performance.

The enhanced understanding provided by the interviewees enabled a sufficient level of support for our findings to formulate action statements rooted in practice. These 'if... then... because' action statements provide those implementing IOCs with a guide on configuring the context in a manner that might promote success [43, 44]. Thus, this may provide a better chance of performance benefits being attained later in the causal chain. Conducting these interviews in England and in its associated regulatory context also allowed us to formulate recommendations for those managing mandated collaborations, which is key given the current policy direction (Table 3). Overall, building trust and faith is essential in the early phases of IOCs to help buffer against potential conflicts during later stages of implementation [45]. Fostering trust and faith is also key in mandated collaborations, however, in these types, formalisation of collaborative requirements through contract can further help manage risk and enable collaborative behaviour. Yet, it is also important to consider that in voluntary arrangements,

over-contractualisation can undermine trust by creating the impression that genuine trust is not valued in their relationship.

Our interviews took place during the COVID-19 pandemic, and as such, it came up frequently as a topic. However, we did not decide to form specific CMOCs relating to COVID-19 as a context, because it was not the original aim of the project to assess collaboration during a pandemic. Nonetheless, it was clear that the pandemic influenced different IOCs in different ways, with it acting as a distraction from collaborative work in some cases, and stimulating new strands of collaboration in others. In particular, in England, the pandemic removed many financial barriers to collaboration and, according to interviewees, often presented a renewed shared vision for IOCs to latch onto, perhaps enhancing relationship-building. However, at the same time, it may have undermined communication by removing the opportunity for informal face-to-face interactions to occur that are key to building trust between actors.

## The findings in context

Two prior realist evaluations [24, 27] likened the link between collaborative functioning and performance to the concept of 'synergy'—which we sought to make more tangible in this study and wider project. From our perspective, it has been more helpful to perceive the output of collaborative functioning as a scale between degrees of competitive and collaborative organisational behaviour depending on the levels of trust, risk tolerance, and faith. These prior realist evaluations also proposed similar concepts to those mechanisms we identified, including "motivation to engage" as an analogue to our notion of 'faith' [24]. Another recent systematic review of reviews sought to determine 'shaping factors' of how cross-sector healthcare collaborations work, but did not utilise a full realist methodology [16]. It identified resources and capabilities (such as organisational capacity), motivation and purpose (including shared vision, unrealistic aims, competing aims, national policies, and commitment), relationships and cultures (including trust, historic relationships, and communication), governance and leadership (decision making and accountability, leadership support), and external factors (geography, social/economic context) as key 'shaping factors'. The review of reviews also connects key aspects together where there was sufficient evidence to do so, in a similar manner as our review [3, 15]. This included connecting trust and communication, and vision and aims with engagement and involvement (similar to our concept of faith). The findings of this review of reviews lends significant supporting evidence for the findings in this realist evaluation, however, our use of realist methods combined with primary data collection further elucidates the causal chain and processes leading to collaborative behaviour.

In this study, the iterative approach to theory-building was one of the key strengths. Adoption of the prior programme theory as both the theoretical and coding framework for analysis allowed for methodical testing of existing mechanisms and CMOCs. Approaching development of such a complex theory in this manner, drawing on 86 articles [3, 8, 15], and utilising primary data collection across a number of case studies, lends significantly more trustworthiness and validity to the final results. Likewise, attaining the perspective of regulators and policymakers shed light on the regulatory context surrounding implementation of these IOCs.

The next steps for this project involve translating these findings into practical tools to support those seeking to deliver IOCs, providing them with a greater understanding of how to configure context to promote successful implementation. These practical tools will be available alongside the full project report to be released in due course (NIHR127430) [46]. The action statements outlined in this paper provide initial guidance for those implementing such arrangements in England and further afield. While some of the findings are applicable mostly to the English setting, such as the recommendations for mandated collaborations, we

anticipate that our findings around core mechanisms such as trust, faith, risk tolerance, etc., are broadly applicable elsewhere.

This practical guidance will be timely, particularly in England, where regulatory reform is set to be introduced in 2022 which will introduce a "*duty to collaborate*" [12]. Our findings demonstrate that those currently implementing IOCs generally do not believe that collaboration can be forced, simply because genuine interpersonal relationships based on mutual trust cannot be forced. This presents questions for government initiatives which seek to mandate these arrangements about how to resolve such issues. Making collaboration more contractually obliged, as discussed in the *confidence* section, would make some of the interpersonal relationship building that would normally be required less important (at least initially) for driving collaborative behaviour (Fig 2). Thus, our theory proposes that mandated IOCs could be initially better supported by greater formalisation through contracts, while, in the longer term, hoping to build genuine interpersonal relationships based on trust upon the foundation of the initial legal structure.

## Limitations

Our sample was entirely focused on the English health system where the healthcare system is largely publicly provided, and it is not clear whether our findings would apply to other contexts with different systems. Our convenience sample also meant that it was likely we had over-representation of IOCs which were performing well. Moreover, we had a significant number of high-level employees within the IOCs but were lacking interviews with frontline workers who may have had a different outlook on IOCs in which they were involved. This recruitment difficulty was partly due to the impact of COVID-19 which occurred shortly after this study began. In cross-sector IOCs, the majority of our interviewees were also from involved acute care organisations, which could have skewed our results, due to the larger size of these organisations and the power dynamics at play. We were also not able to obtain representation of Primary Care Networks and other arrangements unique to England, such as buddying, in our sample, despite our attempts to contact examples of such IOCs.

A further limitation of the analysis was that we were not able to tie aims/drivers of collaboration to outcomes in a manner that allowed us to form explicit CMOCs about them. This is because, when asked, interviewees were largely homogenous in their responses (i.e., all citing similar objectives for their IOCs, such as improving efficiencies, reducing variation in care) regardless of the IOC form. As a result, we focused more on underlying characteristics of IOCs, including whether they were mandated, inter- or intra-sector, or how integrative the forms were, which interacted more directly with various aspects of the programme theory. Due to the focus on the inner workings of collaboration case studies, this analysis leaned more heavily on the practitioner side of the interviews rather than the regulator/policymakers from the sample. Lastly, use of our prior theory in constructing the coding framework could have biased the analysis in favour of identifying excerpts which would support our theory, however, we attempted to mitigate this by being actively aware of this possibility during analysis.

## Conclusion

We conducted a realist evaluation to identify *how*, *why*, and *for whom* inter-organisational collaborations in healthcare work. This evaluation adopted our prior realist programme theory as a middle-range theory, and sought to affirm, refute, refine, and identify new CMOCs by drawing on 32 interviews with practitioners and regulators/policymakers involved with such arrangements and one focus group with patient representatives. Our findings affirm our priory theory; a range of mechanisms were reconfirmed as essential to the functioning of

collaborations, including trust, faith, risk tolerance, confidence in contract, and more. Action statements were formulated to phrase our theoretical findings in a more actionable manner that could be drawn upon by those implementing IOCs in England and elsewhere. Future research will further elucidate how IOCs may improve organisational performance and will aim to further translate these findings into practical guidance for use by those charged with leading and implementing these collaborative arrangements.

## Supporting information

**S1 File. Interview guide.**
(DOCX)

**S2 File. Coding framework.**
(DOCX)

## Acknowledgments

The authors would like to thank the interviewees, advisory group for the study, and patient representatives, without whom this study would not be possible.

## Author Contributions

**Conceptualization:** Justin Avery Aunger, Ross Millar.

**Data curation:** Justin Avery Aunger.

**Formal analysis:** Justin Avery Aunger, Ross Millar.

**Funding acquisition:** Ross Millar, Anne Marie Rafferty, Russell Mannion, Joanne Greenhalgh, Deborah Faulks, Hugh McLeod.

**Investigation:** Justin Avery Aunger, Ross Millar, Anne Marie Rafferty, Deborah Faulks.

**Methodology:** Justin Avery Aunger, Ross Millar, Joanne Greenhalgh.

**Project administration:** Ross Millar.

**Supervision:** Ross Millar, Russell Mannion, Joanne Greenhalgh.

**Validation:** Justin Avery Aunger, Ross Millar.

**Visualization:** Justin Avery Aunger.

**Writing – original draft:** Justin Avery Aunger.

**Writing – review & editing:** Justin Avery Aunger, Ross Millar, Anne Marie Rafferty, Russell Mannion, Joanne Greenhalgh, Deborah Faulks, Hugh McLeod.

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
