## [Decision Letter · Decision Letter 0]

3 Nov 2021

PONE-D-21-29573How, when, and why do inter-organisational collaborations in healthcare work? A realist evaluationPLOS ONE

Dear Dr. Aunger,

Thank you for submitting your manuscript to PLOS ONE. After careful consideration, we feel that it has merit but does not fully meet PLOS ONE’s publication criteria as it currently stands. Therefore, we invite you to submit a revised version of the manuscript that addresses the points raised during the review process.

We look forward to receiving your revised manuscript.

Kind regards,

Ali B. Mahmoud, Ph.D.

Academic Editor

PLOS ONE

Journal Requirements:

2. We note that Figures 1 and 2 in your submission contain copyrighted images. All PLOS content is published under the Creative Commons Attribution License (CC BY 4.0), which means that the manuscript, images, and Supporting Information files will be freely available online, and any third party is permitted to access, download, copy, distribute, and use these materials in any way, even commercially, with proper attribution. For more information, see our copyright guidelines: http://journals.plos.org/plosone/s/licenses-and-copyright.

a. You may seek permission from the original copyright holder of Figure(s) [#] to publish the content specifically under the CC BY 4.0 license.

Reviewers' comments:

Reviewer's Responses to Questions

**Comments to the Author**

1. Is the manuscript technically sound, and do the data support the conclusions?

Reviewer #1: Yes

Reviewer #2: Yes

Reviewer #3: Partly

Reviewer #4: Yes

2. Has the statistical analysis been performed appropriately and rigorously? 

Reviewer #1: N/A

Reviewer #2: N/A

Reviewer #3: N/A

Reviewer #4: N/A

3. Have the authors made all data underlying the findings in their manuscript fully available?

Reviewer #1: No

Reviewer #2: Yes

Reviewer #3: Yes

Reviewer #4: Yes

4. Is the manuscript presented in an intelligible fashion and written in standard English?

Reviewer #1: Yes

Reviewer #2: Yes

Reviewer #3: Yes

Reviewer #4: Yes

5. Review Comments to the Author

Reviewer #1: Thank you for asking me to review this paper, which presents a further development of the authors’ realist investigation into the factors that enable the effectiveness of inter-organisational collaborations in providing health and care services to populations across England.

The paper is clearly written and describes how a realist evaluation methodology has been used to analyse qualitative data from organisational leaders, service users and stakeholders to refine existing CMOCs, and create new ones, building on a programme theory previously published. As such it is original work, and results in the development of a series of plausible action statements, contextualised within relevant contemporary literature on collaboration between public sector organisations. The authors intend to publish these action statements as a practical guide for IOCs to aid their development. Given the increasing requirements for organisations to collaborate in the UK Government’s proposed Health and Care Bill, this can only be helpful. The authors have stated that they are not able to make the original data (interview transcripts) available to preserve the anonymity of the study participants, which is reasonable.

Here are some minor issues that the authors can consider addressing to improve clarity further.

1. Line 100: The authors could reiterate their rationale for claiming that moving from competitive behaviour towards collaboration is ‘ideal’ given the historically contingent nature of this claim. While it's illustrated in the programme theory diagram, this is important because this claim goes on to inform language choices throughout the piece. For example, the authors regard leadership behaviours that are conducive to collaboration as ‘positive’ (line 283) vs those that aren’t which are ‘negative’ (363). Organisations are described as moving ‘back towards competing’ (329) implying regression. The authors could consider either justifying their choice of absolute terms, or using alternative terms that makes their contingency explicit.

2. Line 180 onwards: the authors mention the use of a focus group attended by eight ‘participant representatives’. It’s unclear what a ‘participant representative’ is, and only methods and participant details are described further for interviews. Further information about the focus groups should be included for completeness.

3. Line 244: the link between the quotation and the surrounding text should be clarified – it’s not obvious how it is relevant to the analysis.

4. Line 294: the link between the quotation and the text before it should be clarified – the authors seem to be equating empathetic leadership with support and humility but the connection isn’t obvious.

5. Line 304: given the authors mention that their interviews moved online during the pandemic (200), they could consider whether the significant changes to how people work globally impacts on the finding the communication is adversely affected by geographical difference. Relatedly, to ensure the article is as current as possible, the authors could acknowledge somewhere in the paper how some of the significant changes to working practices that have resulted from the COVID pandemic might impact upon the utility of their action statements.

6. Line 316-317: the authors should consider their use of the term altruistic – typically altruism doesn’t involve getting something back later in the way implied, so a different word might be more appropriate.

7. Line 393: The connection between the quotation and the derivation of the new CMOC could be clarified as the quotation itself could be read as commenting the diversity of the population rather than the particular demographic group identified.

8. Line 462: The authors could explain what they mean by ‘infrastructure’ – the quotation appears to be about the codification of key governance processes for which ‘infrastructure’ seems too broad a term.

9. Line 505-6: the first two sentences under ‘cultural integration’ appear to be inconsistent. If culture is being defined as shared behaviours and attitudes (i.e., an emergent property of social relations), then saying it is ‘essential to organisational success’ doesn’t quite work. They authors should clarify what they mean here.

10. Line 570: this sentence appears to be incomplete.

11. Line 575: the connection between the quotation and the surrounding text is unclear. The quotation appears to describe the relationship between accountability and conflict within an organisation, so it would be helpful to clarify how this fits into the theory for IOCs.

12. Line 593-4: the phrasing of the sentence starting “In England” could be clarified – are the authors talking about a particular model of collaboration that involves privatisation or a more general point? If the former, this could do with a reference.

13. Line 701: the authors could clarify what they mean by “genuine” in relation to interpersonal relationships and be explicit about the qualities they are referring to.

14. Some of the references appear to be incomplete

Reviewer #2: The authors have undertaken a comprehensive examination of interorganisational collaborations. The article is interesting, insightful and particularly timely for IOC stakeholders in England (and likely in other geographies too!). I have provided a few minor suggestions for amendments to a future version of this manuscript in the attached.

Reviewer #3: Thanks for inviting me to review this timely realist review of the factors supporting effective collaboration between health (and care) organisations. With the formation of ICSs in the English NHS this paper is clearly timely for an English audience but it also has wider relevance .

There has been a huge amount of research on integrated care and collaborative working and much is already known about the factors shaping success. This paper covers some of the same ground as other research, but it also highlights some additional themes and mechanisms which are less commonly described in existing literature such as perception of complexity of task and perception of progress and the action statements presented in table 3.

The study also adds to existing knowlege because its methodological approach focuses on the interplay between different success factors and barriers to progress. While it is long, and covers many issues, the table of context/outcome mechanism statements does include some original insights which could be useful for those leading integration work. For example, acknowledging that some people have had a negative experience of integration in the past and this needs to be mitigated.

The research report is presented in line with Rameses II reporting standards and the methods employed and methods of analysis have been well described.

I have some specific comments about the paper – a few of which ask whether there is any additional data to strengthen the points being made – albeit recognising the challenge of word count!

1) There is an imbalance between the various themes presented. For example the long and detailed presentation on Trust is followed by a short section on risk tolerance. Both themes are recognised to be important in shaping success or failure but much less is said about the latter. And, while the authors rightly point out that one or other party in a collaboration has to give some ground, they haven’t touched on other important aspects of risk taking – such as when board members have to approve risks in the interpretation of regulatory constraints or financial controls– without which various past attempts to collaborate have failed. Are there any other data on risk that could add further insights on this topic? The section on capacity felt similarly weak compared to other sections.

2) Lines 286 – 300 provide some good data on the challenges and techniques of good leadership

3) The section on task complexity and scale is interesting and well presented with good use of quotes.

4) The section on culture rang true and seems very pertinent for an English audience given recent mergers of CCGs, and the formation of primary care networks.

5) There is reference in conclusion to regulator perspectives but only one regulator quote is presented (from NMC) and that related to leadership. oIt would be useful to have more data to demonstrate regulators views.

6) Two of the action statements in table 3 are unclear / hard to understand. Specifically, the action point on risk seems to be about trust rather than risk and the second bullet point on managing mandated collaborations.

Overall, I think this paper adds new insights about collaborative working (albeit that some bits could be strengthened ). It is clearly written and meets the standards of the Rameses II guidelines and I recommend publishing it. The editors will need to decide whether to request some additional data to strengthen the sections on risk and on capacity and on regulatory perspectives.

Reviewer #4: Thank you for inviting me this realist evaluation examining 'How, when, and why do inter-organisational collaborations in healthcare work?'

I really enjoyed reading this paper and find it hard to fault!

It is topical with the emergence of ICSs/ICPs and within these primary care networks in the English NHS. It builds nicely on an already published realist synthesis, using well-described realist methods. The paper draws on a wide range of stakeholders for the interviews, although it would have been nice to also see input from stakeholders in Primary Care Networks too.

A couple minor points to help the reader follow what is being presented:

- Just before presenting table 2 it would be helpful for the main text to explain that the ‘novelty’ and ‘affirmation’ relate to the middle range theory from the realist synthesis - currently the words new and existing are used.

- I got a bit confused how the mechanisms listed in the first column of table 2 then relate to the subheadings after. Could this be made clearer? Is there a subheading missing which is in the table? Could the mechanisms and subheadings be numbered?

I look forward to seeing the next paper on practical recommendations.

6. PLOS authors have the option to publish the peer review history of their article (what does this mean?). If published, this will include your full peer review and any attached files.

Reviewer #1: No

Reviewer #2: **Yes: **Stephanie Kumpunen

Reviewer #3: **Yes: **Dr Rebecca Rosen

Reviewer #4: **Yes: **Dr Luisa M Pettigrew

---

## [Author Response · Author response to Decision Letter 0]

15 Nov 2021

Please see the attached response to reviewers file and cover letter.

---

## [Decision Letter · Decision Letter 1]

30 Mar 2022

How, when, and why do inter-organisational collaborations in healthcare work? A realist evaluation

PONE-D-21-29573R1

Dear Dr. Aunger,

We’re pleased to inform you that your manuscript has been judged scientifically suitable for publication and will be formally accepted for publication once it meets all outstanding technical requirements.

Kind regards,

Ali B. Mahmoud, Ph.D.

Academic Editor

PLOS ONE

Additional Editor Comments (optional):

Reviewers' comments:

Reviewer's Responses to Questions

**Comments to the Author**

1. If the authors have adequately addressed your comments raised in a previous round of review and you feel that this manuscript is now acceptable for publication, you may indicate that here to bypass the “Comments to the Author” section, enter your conflict of interest statement in the “Confidential to Editor” section, and submit your "Accept" recommendation.

Reviewer #3: All comments have been addressed

Reviewer #4: All comments have been addressed

2. Is the manuscript technically sound, and do the data support the conclusions?

Reviewer #3: Yes

Reviewer #4: Yes

3. Has the statistical analysis been performed appropriately and rigorously? 

Reviewer #3: N/A

Reviewer #4: N/A

4. Have the authors made all data underlying the findings in their manuscript fully available?

Reviewer #3: Yes

Reviewer #4: Yes

5. Is the manuscript presented in an intelligible fashion and written in standard English?

Reviewer #3: Yes

Reviewer #4: Yes

6. Review Comments to the Author

Reviewer #3: the authors have responded reasonably to comments from all reviewers.

I believe this paper shoudl be published.

Reviewer #4: The paper reads very well and is a very relevant piece of research in the emerging context of ICSs and PCNs.

7. PLOS authors have the option to publish the peer review history of their article (what does this mean?). If published, this will include your full peer review and any attached files.

Reviewer #3: **Yes: **Dr Rebecca Rosen

Reviewer #4: **Yes: **Luisa Pettigrew

---

## [Editor Report · Acceptance letter]

1 Apr 2022

PONE-D-21-29573R1 

How, when, and why do inter-organisational collaborations in healthcare work? A realist evaluation 

Dear Dr. Aunger:

I'm pleased to inform you that your manuscript has been deemed suitable for publication in PLOS ONE. Congratulations! Your manuscript is now with our production department. 

Kind regards, 

on behalf of

Dr. Ali B. Mahmoud 

Academic Editor

PLOS ONE